# "*As it is about that, they do as they please*": Women's experience of the accessibility and acceptability of postabortion care in Kaya, Burkina Faso

Rachidatou Compaoré[1,2]*, Clementine Rossier[3], Onikepe Owolabi[4], Adama Baguiya[1,2], Caron Kim[5], Moussa Zan[6], Nazi Vincent Bagnoa[1,2], Martin Bangha[7], Seni Kouanda[1,2,8]

1 Doctoral School Science, Health and Technology (ED/2ST), Saint Thomas d'Aquin University (USTA), Ouagadougou, Burkina Faso, 2 Biomedical and Public health Department, Institut de Recherche en Sciences de la Santé (IRSS), Centre National de Recherche Scientifique et Technologique (CNRST), Ouagadougou, Burkina Faso, 3 Institute of Demography and Socioeconomics, University of Geneva, Geneva, Switzerland, 4 Guttmacher Institute, New York, New York, United States of America, 5 Department of Sexual and Reproductive Health and Research, UNDP/UNFPA/UNICEF/WHO/World Bank Special Programme of Research, Development and Research Training in Human Reproduction (HRP), World Health Organization (WHO), Geneva, Switzerland, 6 University of Joseph Ki-Zerbo, Ouagadougou, Burkina Faso, 7 African Population and Health Research Center APHRC, Nairobi, Kenya, 8 African Institute of Public Health, Ouagadougou, Burkina Faso

* rachidoc7@gmail.com

## Abstract

Accessing acceptable postabortion care (PAfor remains a major challenge in sub-Saharan Africa, where abortion is legally restricted and highly stigmatized, particularly for young and socioeconomically disadvantaged women. Evidence on the specific obstacles faced by these groups in rural and semi-urban settings is limited. This study explored women's experiences with PAC quality in Kaya, a small town in Burkina Faso. We conducted in-depth interviews with women (n = 35) who had participated in a respondent-driven sampling survey on abortion safety in 2021 and had sought PAC for abortion-related complications in formal health facilities. Additional interviews were carried out with representatives of local grassroots associations involved in sexual and reproductive health in the town (n = 8). Thematic analysis was used to examine perceptions of abortion stigma within the community, followed by a typology analysis to identify inequalities in the accessibility (timely, geographically reasonable, and affordable provision of care) and acceptability (patient-centeredness, including cleanliness, privacy, respect and non-judgemental care) of PAC across different categories of women. Both women and association representatives strongly condemned abortion, and the legal exceptions were largely unknown. Abortion stigma was much stronger for young and unmarried women who often sought abortions due to partners denying responsibility and fear of social exclusion. Married women faced less stigma when abortions were linked to closely spaced pregnancies or economic hardship. Fear of prosecution and mistreatment delayed care-seeking, especially

**Data availability statement:** Due to ethical concerns related to participant confidentiality, the data supporting the findings of this study cannot be made publicly available. This qualitative research was conducted in a geographically limited Health and Demographic Surveillance System (HDSS) site in Burkina Faso, where abortion remains highly stigmatized. Given the sensitive nature of the topic and the detailed profiles of participants, even anonymized data could lead to deductive disclosure and compromise participant privacy. The study protocol was reviewed and approved by the Comité d'Éthique pour la Recherche en Santé (CERS) of Burkina Faso, under the condition that data access would be restricted to the research team. Participants were explicitly informed that their data would not be publicly shared or deposited in repositories. For inquiries regarding the ethical restrictions on data sharing, please contact the current president of the CERS at the following email address: kouetafla@yahoo.com We remain committed to responsible data sharing and are open to considering individual data requests, which may be submitted to the corresponding author and will be reviewed on a case-by-case basis, provided that participant confidentiality can be fully preserved.

**Funding:** RC received funding from the HRP Alliance, part of the UNDP-UNFPA-UNICEF-WHO-World Bank Special Programme of Research, Development and Research Training in Human Reproduction (HRP), a co-sponsored program executed by the World Health Organization (WHO), through the LID HUB program (WHO PO no. 2017/740863). This article represents the views of the named authors only, and does not represent the views of the World Health Organization. The funders had no role in study design, data collection and analysis, decision to publish, or preparation of the manuscript.

**Competing interests:** The authors have declared that no competing interests exist.

among unmarried adolescents and young women. They also reported more verbal abuse, longer waiting times, and a lack of privacy during care, reflecting and reinforcing the heightened stigma they faced. Women with greater socioeconomic resources more often used private facilities, where both accessibility and acceptability were perceived as better, particularly for unmarried women. This study highlights substantial inequalities in the accessibility and acceptability of PAC in Kaya, shaped by women's marital status, age, and socioeconomic position.

## Introduction

Unsafe abortions pose a significant public health concern globally, particularly in sub-Saharan Africa, which reported the highest abortion-related case-fatality rate in 2019, with approximately 185 maternal deaths per 100,000 abortions [1,2]. Despite being recognized as a human right, access to safe abortion remains limited due to various unfavorable social, cultural, and legal factors [2]. Unsafe abortions can lead to severe maternal complications and mortality, emphasizing the importance of prompt and quality postabortion care (PAC) in healthcare facilities. [3].

PAC, a global strategy initiated in the early 1990s, comprises five essential elements: emergency treatment of incomplete and unsafe abortions, provision of contraceptive and family planning services, community and service provider partnerships to prevent unwanted pregnancies and unsafe abortions, mobilization of resources for timely care, and ensuring that health services meet community expectations and needs [4]. Although progress has been made in providing PAC services, obstacles still prevent individuals with low socioeconomic status and those living in rural areas from accessing high-quality care. These barriers result in significant disparities in the utilization of PAC services and create significant health inequalities [5]. Several factors contribute to these barriers, including limited accessibility, inadequate infrastructure, provider attitudes and skills, financial constraints, lack of legal knowledge, and fear of stigma [5–7].

Abortion care quality is determined by safety, effectiveness, accessibility, patient-centeredness, acceptability, timeliness, efficiency, and equity, as recognized by the World Health Organization (WHO) and the Institute of Medicine (IOM) [8–10]. Recent efforts have focused on promoting the quality of abortion and PAC to reduce maternal morbidity and mortality [10]. However, there is still a lack of consistency in measuring the quality of abortion care, particularly in terms of interpersonal aspects., which encompasses women's experiences and perceptions of healthcare interactions [10–12].

Studies conducted in sub-Saharan Africa have demonstrated that women in the region value interpersonal factors such as trust, empathy, respect, support, confidentiality, and nonjudgmental care when seeking abortion-related services, similar to women in other settings [11–15]. Providing care that aligns with individual needs and preferences, or patient-centered care, is thus important in ensuring quality of care in this region, as in other regions. Despite its proven benefits, a recent review revealed

that this dimension of quality of care is often overlooked in the provision of abortion services [16]. Furthermore, there are significant disparities in the interpersonal aspects of quality care, with adolescents and women of low socioeconomic status experiencing disrespect, discrimination, and provider bias, leading to delays in seeking care [17–20]. Interactions with healthcare providers significantly influence women's perceptions of quality, care-seeking behavior, and adherence to treatment recommendations [21–23].

Existing research on the quality of abortion care in sub-Saharan Africa mainly focuses on urban areas with high abortion rates and legal contexts, with limited information available on rural areas and small towns [8–10]. This is concerning since a significant proportion of induced abortion cases in rural areas require treatment, putting poor women at a higher risk of complications [2]. Moreover, the influence of social expectations and abortion stigma on women's acceptance of PAC remains unclear.

In Burkina Faso, abortion is legally permitted under specific circumstances, as outlined in the 2018 Penal Code. A doctor must certify that continuing the pregnancy poses a threat to the woman's health or that the unborn child is likely to suffer from a serious disease or condition that cannot be cured at the time of diagnosis. Additionally, the law allows for the termination of pregnancy due to rape or incest if the distress is established by the public prosecutor within the first 14 weeks. However, those who violate this law or attempt to do so are subject to punishment [11]. Approximately one-third of Burkinabe women are aware that abortion is legal, and they tend to be younger, better educated, live in urban areas, be unmarried, and be without children [12]. Voluntary abortion is not only illegal but also faces strong social and religious disapproval in the country, as human life is considered sacred from the moment of conception, and abortion is generally viewed as a consequence of immoral behavior, such as extramarital sex. The stigma attached to abortion leads women to seek it discreetly, fearing prosecution and social stigma. Clandestine abortions are often performed under unsafe conditions, posing a risk to women's health and lives [13]. In 2020, complications from unsafe abortions were the fourth leading cause of maternal mortality in Burkina Faso [13]. According to the PMA abortion survey results in the country in 2022, over half of abortions were unsafe (51%), and there is evidence of disparities among older women, women with less education, and women living in rural areas [14]. PAC is an integral part of the national free care policy and falls under the Emergency Obstetric and Newborn Care (EmONC) framework [15]. However, despite these policies, only half of the women who reported a potential severe abortion complication accessed PAC at a facility for treatment [14].

Previous studies have reported that acceptability (i.e., person-centeredness, the dimension of quality of care that interests us the most) depends on varied social and practical norms anchored in gender dynamics and power relationships that are specific to each context [9,16,17]. Hence, Olivier de Sardan, in a study on maternal and childcare in West Africa, highlighted how the standard interventions or "traveling models" developed by international experts to improve aspects of maternal health systems in LMICs, which are usually introduced in an almost identical format in many countries, could fail when implemented in very different contexts by not considering the cultural dimensions of acceptability specific to the local context [18]. While several studies have provided an overall assessment of abortion frequency and safety in Burkina Faso, we have limited information on women's experiences with PAC quality at the facility level, especially when considering patient-centeredness or acceptability.

Our study aimed to understand how individual and contextual cultural factors influence women's experiences of accessing acceptable PAC. We conducted a survey in Kaya, a secondary city in Burkina Faso, using respondent-driven sampling (RDS) at the population level to identify participants [19] who had experienced induced abortion. Our data collection tool incorporated questions from both the WHO framework for quality of care [20] and the abortion stigma literature [9,24–26]. We examined attitudes toward abortion among abortion seekers and representatives of sexual and reproductive health (SRH) associations, identified stigmatized and less-stigmatized types of abortions, and explored the relationships between abortion stigma, accessibility to the PAC, and the acceptability of interpersonal relations during the PAC. Our research hypothesized that understanding the cultural context and client perspectives is crucial for improving the accessibility and acceptability, or person-centeredness of PAC. This approach supports equitable access to quality care for abortion

complications in Burkina Faso's restrictive setting, contributing to SDG 3.7 and SDG 5.6 on universal access to sexual and reproductive health services and rights [27].

## Materials and methods

### Study site

The Kaya Health and Demographic Surveillance System (Kaya HDSS) is a research platform located in the Centre-North region of Burkina Faso in the Kaya Health District, and was created in 2007 by the *Institut de Recherche en Sciences de la Santé* (IRSS). The site was 70% urban and 30% rural [28]. The population of the Kaya HDSS is estimated to be 75,330 inhabitants in 2021, 19,622 (26.0%) of whom are women aged 15 years and older [13].

The Kaya health district includes forty (40) first-level public health facilities, including 39 primary health and social promotion centers (CSPSs) and one medical center with a surgery unit (CMA). In addition, there are four private and confessional health facilities. The health district includes the regional hospital of Kaya (CHR), which is a referral hospital for the Centre-north region. While some CSPSs offer basic care, comprehensive PAC services, including the management of abortion complications, are primarily concentrated in higher-level facilities such as CMA and CHR. In 2020, 53 obstetric complications due to unsafe abortions were reported in the Kaya Health District, representing 8.3% of all obstetric complications [13].

Kaya, the capital of the Centre-North region, has been facing a massive influx of internally displaced persons fleeing terrorist attacks in the north of the country and has been the target of terrorist attacks by jihadist movements since 2015. This situation has created another layer of complications, where migrants face increased gender-based violence, sexual violence, and greater difficulties in accessing SRH services.

### Study design, study population and participant selection

We conducted a qualitative study using semi-structured interviews with key informants between October and December 2021, including women who underwent an induced abortion outside the cases authorized by law and had a history of PAC. The latter were identified among participants in an RDS survey conducted from August to November 2021 at the surveillance site, which gathered quantitative data on 481 recent abortion seekers to measure abortion safety and was also conducted in the slums of Nairobi, Kenya [19]. This approach was particularly helpful in collecting information on informal sector abortion, as it is difficult to identify abortion seekers in usual population-based sampling because of the secrecy associated with the practice [29,30]. The eligibility criteria for RDS participants included being aged 15–49 years, living in the surveillance area, and having undergone an induced abortion within the past three years. The full process of the RDS survey has been described previously [31]. At the end of the RDS interviews, 89 (18.28%) of the 487 abortion cases had complications, and 68 (76.40%) of these complicated cases sought care at the facility level. Women who reported complications from induced abortion and had used health services for PAC were invited to participate in this qualitative survey. Thirty-five women agreed to participate in the interviews, corresponding to an acceptance rate of 51.47%.

The key informants also included representatives from local associations (N = 8). Purposive sampling was used to select eight local associations whose activities included community awareness of sexual and reproductive health (SRH) in the Kaya HDSS area. The head of the association or a representative at the local level was invited to participate. All of these associations have been working on SRH in the area for more than 20 years, often in collaboration with the health district. The purpose of the interviews with these stakeholders was to describe the general social and cultural context surrounding abortion in Kaya to situate the discourses of women seeking abortions.

### Data collection and treatment

Data from women were collected by experienced female research assistants who were initially involved in the RDS survey and fluent in both Moore and French, the two most used languages at the study site. In addition to their prior training

in RDS surveys and the ethical aspects of data collection on sensitive issue-induced abortion and stigma, the research assistants underwent two additional days of booster and reinforcement training on qualitative interviews, which included role-plays and exercises on probing, neutrality, and managing emotional responses.

Individual interviews with women with an abortion history were typically conducted outside the participant's home in a private location and at a time of their choosing. Each participant was interviewed in the language of her choice (Moore or French) by a single research assistant, without additional personnel present. Given the sensitivity of the topic and participants' discomfort and refusal to be recorded, the interviews were not voice-recorded. Instead, each research assistant took real-time notes, primarily in French, with occasional use of Moore to capture culturally specific expressions. In most cases, the transcripts and field notes were written simultaneously during the interview. Immediately afterward, the interviewers expanded their notes to ensure completeness and integrate contextual and nonverbal observations. The expanded transcripts were then translated into French when necessary, ensuring that the local meanings and nuances were preserved.

The Interpersonal Quality in Abortion Care scale was used to construct the section on patient-centered care in our interview guide [32]. The questions focused on the women's knowledge and perception of abortion and related laws, the pathways to induce abortion and access PAC, their experience of care, and the stigma perceived during informal sector abortion and formal postabortion services. Interpersonal relations during care were explored according to the following dimensions: information provided, respect and dignity, information, privacy and confidentiality, non-judgment, equity when providing care, and patients' overall satisfaction with the care provided.

Interviews with representatives of local associations were conducted at their organizational headquarters by the same research assistants. Data were collected using interview guides on their perceptions of abortion in their community, knowledge of and position on the law, and perceptions of PAC and PAC quality in the district. The interviews were also conducted in the respondents' preferred language, either Moore or French. These interviews lasted between one and two hours and were recorded with the respondents' consent. The recorded interviews were then translated into French, if necessary, before being transcribed.

All transcripts were typed using Microsoft Word (version 2310). Throughout the data collection process, regular debriefings were held between the research assistants and the core research team. These sessions served to reflect on the interview dynamics, identify emerging themes, and iteratively adapt the interview guide. For example, as informal care pathways and legal misconceptions emerged, new probes were added to further explore these areas.

## Data analysis

We conducted a thematic analysis using Braun and Clarke's reflexive approach, which emphasizes the researcher's active role in interpreting patterns of meaning within qualitative data [33]. This process involved six iterative phases: familiarization, initial coding, theme development, theme refinement, theme definition, and final reporting [33]. The lead author, fluent in both Moore and French, performed all coding in French, drawing on her bilingual and contextual expertise to ensure that the interpretations were culturally grounded.

The initial codebook was informed by the existing literature on abortion stigma and patient experience in abortion and PAC contexts [9,10,20,34–36], with additional inductive codes developed as themes emerged from the dataset. The analysis focused on two core dimensions of the WHO's Quality of Care framework: accessibility and acceptability/patient-centeredness (Fig 1) [20]. Accessibility refers to timely, geographically reasonable, and affordable provision of care. All inputs should be accessible within a safe physical reach for all populations, with an emergency care function of 24 h/7 days and free or affordable health services based on non-discrimination and equity. Acceptability or patient-centeredness includes patient preferences and aspirations that should be considered in the healthcare provided. Care should be patient-centered, respectful of the patient's values and choices to promote patient satisfaction and fulfillment of human rights. Moreover, the environment should be safe, clean, appealing, and designed to respect the privacy and confidentiality of those seeking such services. Healthcare providers should be non-judgmental, considerate, and easy to relate to. The

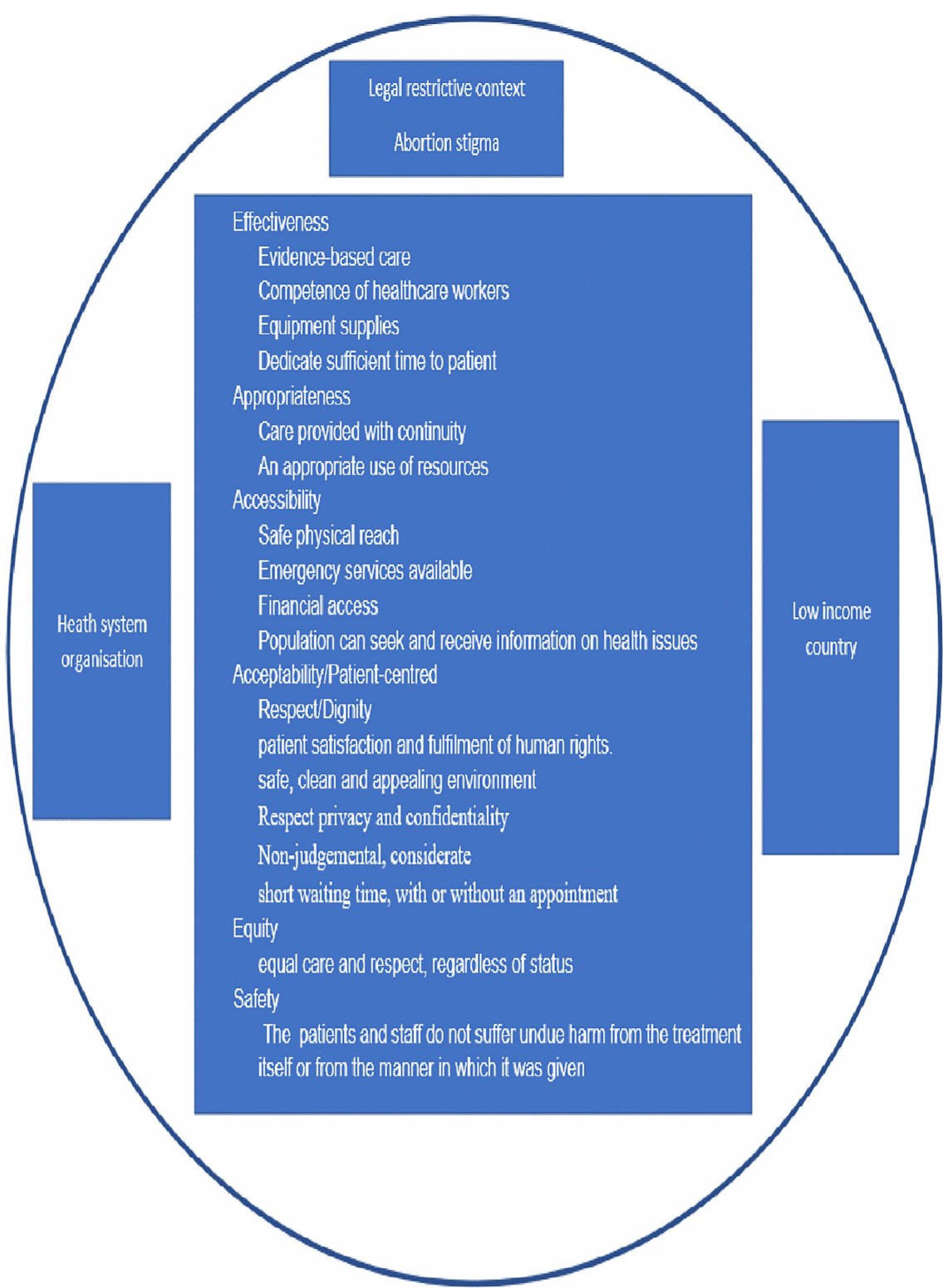

**Fig 1. Conceptual framework for PAC quality care from women's perspective.**

facility ensures that a triage system is in place, and that consultations occur within a short waiting time, with or without an appointment, and (where necessary) a swift referral is organized [20]. Abortion stigma was assessed through respondents' perceptions of community attitudes toward this practice, their knowledge of the law, and how women view their own abortions in light of prevailing social expectations.

Interpretive depth was enhanced through regular debriefings between the lead author and the research team that supervised the RDS study, who shared reflections from the fieldwork that shaped the thematic focus and contextual alignment. Additionally, peer consultations with the co-authors supported the critique and refinement of thematic categories throughout the analysis.

Although coding was conducted by a single researcher, reflexivity was rigorously maintained through analytic memoing, ongoing documentation of interpretive decisions, and structured reflection on the author's positionality. These reflexive practices, alongside a collaborative review of emerging findings, mitigated the limitations associated with single-coder analysis and strengthened the study's trustworthiness.

To further link macro-level cultural norms with individual care experiences, we developed two typologies based on women's socioeconomic status and the degree of stigma surrounding their abortion. These typologies provide a framework for analyzing variations in PAC trajectories and barriers to acceptable care. All data were analyzed using ATLAS.ti software (version 8.4.26.0, 2019).

This study adhered to the Standard for Reporting Qualitative Research (SRQR) [30], that is provided as a supplementary file (S1 File).

### Ethical consideration

This study was approved by the National Ethics Committee for Research in Health of Burkina Faso (no. 2020-02-033). Participation in this study was voluntary. Verbal informed consent was obtained from each respondent before conducting the interviews. For participants under the age of 15, parental consent was obtained before seeking verbal assent from the participant. This two-step procedure ensured compliance with national ethical regulations and safeguarded the rights of minors participating in research related to sensitive topics. Anonymity and confidentiality were guaranteed in the analysis and subsequent reporting of the study results by removing all identifiers or personal information.

## Results

In this section, we first present the participants' perceptions of abortion in their community and their knowledge of the law. We then examine the reasons women give for their abortion and their perceptions of its social acceptability, which mirrors prevailing expectations in their cultural context. We then classified women into different groups based on having more or less acceptable abortions and socioeconomic means. Finally, women's experiences of accessing PAC and their perceptions of the acceptability and person-centeredness of the PAC they received were reported across groups.

### Sociodemographic characteristics of study participants

The study included 35 women who sought PAC in Kaya and eight community representatives from local SRH associations.

The characteristics of the women with PAC experience are presented in Table 1 (Table 1).

Most participants were under 24 years old (71.4%), unmarried (62.9%), had a secondary school education or less (94.3%), and had no children (57.1%). Most of them had not had an abortion before (82.9%), and approximately two-thirds of the abortions were considered unsafe (62.9%), meaning they were performed outside a health facility or with a method not recommended by a qualified provider (medical doctor, midwife, or trained nurse). All women lived in the semi-urban area of Kaya, and 85.0% were Muslim. Only one participant had worked in the formal sector, while the rest were students, housewives, or worked in the informal sector.

**Table 1. Characteristics of RDS respondents re-interviewed in this study.**

| | Adolescents and youth(15-24y) (n = 25) (%) | | Women aged 25 and plus (n = 10) (%) | Total (n = 35) (%) |
|---|---|---|---|---|
| | Adolescents (15-19y) 10(40.0) | Youth (20-24y) 15(60.0) | | |
| **SOCIODEMOGRAPHICS** | | | | |
| **Marital status** | | | | |
| Single | 10(100.0) | 11(73.3) | 1(10.0) | 22(62.9) |
| Married | 0(0.0) | 2(13.3) | 7(70.0) | 9(25.7) |
| Divorce/Widow | 0(0.0) | 2(13.3) | 2(20.0) | 4(11.4) |
| **Education level** | | | | |
| None | 0(0.0) | 0(0.0) | 3(30.0) | 3(8.6) |
| Primary | 0(0.0.0) | 5(33.3) | 4(40.0) | 9(25.7) |
| Secondary | 9(90.0) | 9(60.0) | 3(30.0) | 21(60.0) |
| Tertiary | 1(10.0) | 1(6.7) | 0(0.0) | 2(5.7) |
| **Profession*** | | | | |
| Housewife | 1(10.0) | 3(20.0) | 3(30.0) | 7(20.0) |
| Student | 8(80.0) | 6(40.0) | 1(10.0) | 15(42.9) |
| Informal | 1(10.0) | 6(40.0) | 5(50.0) | 12(34.3) |
| Formal | 0(0.0) | 0(0.0) | 1(10.0) | 1(2.9) |
| **Religion** | | | | |
| Muslim | 10(0.0) | 15(100.0) | 7(70.0) | 32(91.4) |
| Christian | 0(0.0) | 0(0.0) | 3(30.0) | 3(8.6) |
| **Number of deliveries** | | | | |
| 0 | 10(0.0) | 9(60.0) | 1(10.0) | 20(57.1) |
| ≥1 | 0(0.0) | 6(40.0) | 9(90.0) | 15(42.9) |
| **Number of children** | | | | |
| 0 | 10(0.0) | 9(60.0) | 1(10.0) | 20(57.1) |
| ≥1 | 0(0.0) | 6(40.0) | 9(90.0) | 15(42.9) |
| **Number of previous abortion** | | | | |
| 0 | 7(0.0) | 14(93.3) | 8(80.0) | 29(82.9) |
| ≥1 | 3(0.0) | 1(6.7) | 2(20.0) | 6(17.1) |
| **SAFETY OF INDUCED ABORTION** | | | | |
| Safe | 4(40.0) | 3(20.0) | 6(60.0) | 13(37.1) |
| Unsafe | 6(60.0) | 12(80.0) | 4(40.0) | 22(62.9) |
| **TYPE OF FACILITY FOR POSTABORTION CARE** | | | | |
| Private | 3(30.0) | 0(0.0) | 1(10.0) | 4(11.4) |
| Public | 7(70.0) | 15(100.0) | 9(90.0) | 35(88.6) |

* Informal refers to occupations without formal contracts or social protection (e.g., petty trade, domestic work, and casual labor). Formal refers to occupations with formal contracts and social protection (e.g., salaried employment in government or private institutions).

Of the eight representatives from local SRH associations in Kaya, seven were men and one was a woman, with ages ranging from 28 to 50 years. All were married, and their length of service in the associations spanned from 2 to 21 years. Their demographic profiles are summarized in Table 2 (Table 2).

The in-depth interviews for both categories of respondents ranged from 30 to 120 minutes (with a median of 60 minutes).

Participant narratives revealed two predominant demographic profiles among abortion seekers; each associated with distinct abortion-related experiences. To facilitate a comparative understanding of the perspectives explored subsequently, Table 3 provides a synthesis of the main differences and similarities between the two groups (Table 3).

**Table 2. Demographics of community representatives.**

| | Association representatives (n = 10) (%) |
|---|---|
| **SOCIODEMOGRAPHICS** | |
| **Gender** | |
| Female | 1 (12.0) |
| Male | 7 (88.0) |
| **Age group (years)** | |
| 25–34 | 3 (38.0) |
| 35–44 | 3 (38.0) |
| 45–54 | 2 (25.0) |
| **Marital status** | |
| Single | 0(00.0) |
| Married | 8(100.0) |
| Divorce/Widow | 0(00.0) |
| **Education level** | |
| None | 0(00.0) |
| Primary | 2(25.0) |
| Secondary | 5(63.0) |
| Tertiary | 1(12.0) |
| **Primary role** | |
| Association coordinator | 5 (63.0) |
| Regional supervisor | 1 (12.0) |
| Project manager | 2 (25.0) |
| **Years of service** | |
| < 10 | 2 (25.0) |
| 10–19 | 4 (50.0) |
| ≥ 20 | 2 (25.0) |

Given the absence of systematic data on key socioeconomic indicators, such as financial means (including partner support) and social connections (including ties to health workers), we used the type of health facility accessed for post-abortion care (private versus public) as a proxy for the socioeconomic status. According to the available variables (Table 1), single young women aged less than 25 years had a high and homogenous level of education and were often students. Older women have a more heterogeneous social background (according to their level of education and occupation). In the reminder analysis, we examine women's perceptions of the accessibility and acceptability/person-centeredness of PAC, distinguishing the different groups of abortion seekers identified thus far (unmarried versus married, women using public versus private facilities).

## Participants' reasons for abortion

All the women interviewed reported that they had no other choice but to terminate their pregnancies due to their personal living conditions and the challenging context at the time.

First, unmarried young women who are not committed to their partners and who are more likely to be condemned as having low sexual morals. Reasons for seeking abortion in this group are related to denial of paternity, avoiding social shame, and desire to continue their education.

**Table 3. Contrasting experiences of postabortion care across key sociodemographic groups.**

| Dimension | Young, unmarried women (Under 24) | Older, married women (25+) |
|---|---|---|
| Sociodemographic profile | Single, adolescent/young adult, secondary education or less, no children | Married, varied education, typically with children |
| Reasons for abortion | - Denial of paternity<br>- Social stigma<br>- Desire to continue education | - Closely spaced pregnancies<br>- Financial constraints<br>- Unintended pregnancies |
| Stigma perception | High (linked to morality, youth, and lack of marriage) | Moderate (seen as "understandable" or "necessitated") |
| Care-seeking delays | More frequent (due to fear of legal sanction, provider judgment, and shame) | Less frequent (more empowered by social roles and networks) |
| Treatment at facility | Reports of verbal abuse, judgmental attitudes, lack of privacy | Reports of respectful or neutral treatment |
| Facility use | Public (more barriers) or private if socially/financially supported | Both public and private facilities used confidently |
| Accessibility and acceptability | Lower (greater barriers across all dimensions) | Higher (more seamless navigation of care system) |

"*Here, a girl who becomes pregnant without being married is frowned upon by society, and if the author of the pregnancy accepts [to recognize the pregnancy and engage in the union process], so much the better, but if he refuses, the girl is obliged to have a clandestine abortion...*" 19, female, single, student.

Another young girl expressed the same motivation for her abortion:

"*Often, you get pregnant by accident, and you have no choice but to have an abortion. If you keep it (pregnancy), living conditions are not easy. You are taken care of [dependent]. How are you going to take care of yourself since you do not have the means, and the person who got you pregnant refused (to help)?* " 20, female, single, student.

The second group comprised married and older women. The reasons for abortion were different from the first group and were generally seen as more acceptable, with women presenting their situations as accidental.

In this group, women may have wanted to avoid having two pregnancies that were too close together, which can negatively impact the health of both the mother and children and be considered socially undesirable, as stated by this woman:

"*As for my case, the child is small, and I fell pregnant; if one had not helped me to remove it, the pregnancy could kill me, and thereafter, my child was also going to follow... Often it is involuntary when one falls pregnant, there*!" 33, female, married, housewife.

"*In my case, for example, my child was small, so my husband was not supposed to approach me, so he did it, and we were in trouble…It was accidental*" 30, female, married, seller.

Another reason within this category was the financial limitations that impeded women's ability to provide care for an additional child, compelling them to resort to a course of action that they had never envisioned themselves pursuing.

An atypical case was found in a woman who was internally displaced due to terrorism and was a victim of rape.

"*To tell you the truth, I never imagined in my whole life that I would be in such a situation, but life circumstances (displacement and rape) have brought me to this point, …I truly did not have a choice.*" 35, female, widow, housewife.

**"Abortion is a crime": perceptions of abortion**

All respondents reported that abortion was highly stigmatized and largely condemned in their community. Most women who participated in this study, irrespective of their sociodemographic characteristics, described abortion as 'a bad thing' and even a crime, as follows:

*"I know that abortion is not a good thing..."* 36, female, married, informal sector

This negative value attributed to abortion made women feel ashamed of their acts in relation to their community's view:

*"When I think of the crime I committed, I only cry."* 20, female, single, informal sector

*"The real problem is the people around. It is very shameful, and you even want to disappear."* 23, female, single, student

Respondents believed that fear of stigma discouraged women who had an abortion from seeking timely care at health facilities for complications after an induced abortion.

*"… A girl who has had an abortion does not see herself going to the hospital for care because of the shame and the judgment people will have on her.*" 20, woman, single, student.

Association representatives share the same moral stance and condemnation of abortion, often evoking the sacrality of human life from conception at the fetal stage.

*"You may have difficulties, but there is no reason for me that a woman should even think about abortion for a moment. I believe that life is sacred from the moment of conception. It is not even that I think I am convinced that life is sacred. It is so sacred that it should not be trivialized."* 50, male, married, association representative.

Within community narratives, abortion was often linked to perceptions of young girls' premarital sexuality and libertinism, both of which were considered immoral in this context. According to these association representatives, women who abort should be punished, both because of the crime they have committed, and as a warning to those who would think of having an abortion.

*"… We live in societies with global rules. Therefore, your rights are not totally outside the rights of society…We should punish her for trying to have an abortion.*" 50, male, married, association representative.

The practice of abortion was viewed as "imported" and not as part of African culture.

*"…This (abortion) is not part of African culture. It is an imported idea."* 50, male, married, association representative.

Representatives from the association stated that they would advise girls facing unintended pregnancies to carry them to term, as they could face a high risk of complications, including infertility and death.

*"… We think in the first place that it (abortion) is not a good thing because… there is a question of death, it can lead to death; it can also involve the sterility of the person if it (abortion) is not well done."* 48, female, married, representative of the association.

***"I don't know the law as such"*: knowledge of the exceptions in the law**

Both the women and the associations' representatives had limited knowledge of the law surrounding abortion. While legal restrictions on abortion are often known, little is understood about the circumstances under which the law permits the procedure. All the women consistently reported that abortion is forbidden in Burkina Faso and that those involved, including the woman, supporters, and healthcare providers, risk prosecution and imprisonment.

Young single women usually learn about the legal aspects of abortion through informal channels, such as conversations with peers at school or in their neighborhoods.

*"All I know is that if you are caught having an abortion, you will go to prison, you and all those involved… I received this information at school with friends and through talks in the neighborhood."* 19, female, single, student.

Some participants had firsthand experience with legal restrictions.

*"I do not know the law about abortion, if it is not said that when you are caught and you have had an abortion, you can go to prison, that is all; and I myself almost went there because it was my darling who filed a complaint against me to the police, but it was settled amicably between the two families, which avoided that; otherwise, you were not even going to chat with me."* 20, female, single, informal worker.

Older women also reported learning about the abortion ban through informal discussions or radio programs discussing the subject, as reported below:

*"I do not know what it (the law) is, but I know that if you have an abortion and you are caught, it is serious; all those who accompanied you or helped you will go to prison, and it is even more serious if the 'owner of the pregnancy' does not know about it… I know a girl and her aunt who are in prison for this very thing."* 30, female, married, informal worker.

*Yes, I know that the law prohibits induced abortions. I received this information during a broadcast on a radio station and in the neighborhood during talks.*" 34, female, married, informal worker.

Some women stated having learned about the law for the first-time during PAC at the hospital.

*"No, I did not know this law before coming here (in the hospital)…"* 17, female, single, student.

However, it appears that even some women whose cases fell within the exceptions of the law were treated in health facilities as if they benefited from a favor and not a right, and without being informed of that right. This woman, who was forced to flee her village because of insecurity and was the victim of rape, reported the following:

*"… If it is done in secret, it is because it is not authorized… We fled our localities, leaving everything behind, and if we resort to this practice, it is in spite of ourselves because many of us were victims of sexual violence, and it is as a result of this that these pregnancies have occurred."* 35, female, widowed, housewife.

Among association representatives, only two individuals identified rape and incest as exceptions for accessing legal abortions, which they learned through media and professional workshops. No participant reported consulting with a healthcare professional or district officer regarding community actions on unsafe abortions or PAC. Most participants were unfamiliar with PAC and its importance.

*"I cannot tell you that I know the law governing abortion because I cannot even tell you anything about it, but I hear about it. This is through the media. We follow the TV…there are discussions that are organized around this theme that we follow from time to time with controversial points of view, which allows us to know a little bit about what is happening on a national level around the issue."* 50, male, married, association representative.

**"You are afraid to even go to the hospital". Women's experience accessing care**

Most respondents indicated that geographical distance was not a hindrance to accessing PAC. However, concerns about stigma from healthcare providers, limited resources, and fear of disclosing their situation within the community prevented some women from seeking timely care. Participants described varying experiences of care-seeking shaped by factors such as age, marital status, financial resources, social support as well as the type of facility used.

Several unmarried adolescents and young women described delaying care-seeking due to the fear of legal prosecution and/or mistreatment from health providers. These women typically sought PAC in health facilities when they had no other choice or felt their lives were in danger, as reported by one participant.

*"It is not easy when you had an abortion and then there are complications; you're afraid to even go to the hospital for fear of what they're going to tell you or do. It is when you truly do not have a choice, when you feel like you are dying, that you go (to the hospital)."* 19, female, single, student.

Another added:

*When it got hot and the blood was flowing a lot, I felt dizzy, and I had no choice but to go to the hospital. The CSPS is not far from where I live. They took me on a motorbike and sent me without knowing because I fainted afterwards."* 19, female, single, student.

However, adolescents and young women with financial support from a partner or an older relative with connections at a health facility could more easily access PAC services in both public and private facilities.

*"It is like I told you, since I went through someone who already knew the provider, I did not suffer at all; otherwise, I know it is not easy in any case."* 19, female, single, student.

Most women aged 25 or older in this study sought help from their social networks or healthcare workers. They were more likely to seek PAC at healthcare facilities, and reported receiving medical treatment for complications without any negative attitudes upon arrival. Some participants even received treatment from the same healthcare worker who had initially assisted them with the abortion.

*"I did the abortion in the evening, and it was at night approximately 8 p.m. that it started to hurt, so I had to come back to the hospital. My sister called the nurse, and we met at the CSPS. When he arrived, as he knew what it was about, he examined me directly'."* 30, female, married, trader.

*"When I needed care, I went to see a midwife who works at the CSPS and for whom I do the laundry every time. I was lucky because she was on duty and she looked after me nicely."* 33, female, married, laundress.

*"I think that PAC is easy to access; as soon as I arrived, I received my care.... I did not have any difficulty accessing care because, when I arrived, I was received in a room and examined."* 35, female, married, housewife.

Access to PAC was consistent among elderly women, regardless of whether they received care from public or private facilities.

> "*I do not know if it is because I went to a clinic, but I find that PAC is easily accessible. I did not have any problems accessing care, but when I came back and I was in real pain, I called the clinician first, and he asked me to come back to the clinic. I did not have any major difficulties.*" 30, female, married, civil servant.

**"As it is about that, they do as they please": women's perceptions of acceptability/person centeredness in PAC**

Study participants reported their experiences at the healthcare facility where they sought treatment for an abortion complication, focusing on the facility's safety and cleanliness, the interpersonal relationships between patients and caregivers, and their level of satisfaction with the PAC provided.

Participants' reflections revealed that the acceptability and person-centeredness of PAC were deeply influenced by their perceived social position and the discretion of health workers. Young women, particularly those seeking care in public facilities, reported that their treatment often depended on the provider's attitudes toward abortion rather than on standardized care practices. This perceived arbitrariness left participants feeling powerless and judged, and many of them expressed their frustration with what they perceived as health workers' discretionary control over the quality and tone of care in abortion-related contexts.

One respondent captured this sentiment when she said:

> *I do not know what to say about the quality of care I received. As it is about that, they do as they please.*" 21, female, single, informal worker.

For many young women, expectations of respectful and dignified treatment were low, and satisfaction was sometimes defined simply by survival or avoidance of overt humiliation. These lowered expectations further influenced how participants evaluated their post-abortion care, as explored below.

**Safe, clean, and appealing environment and costs of care**

Despite the critical condition of some of the participants at their admission at the health facility, the women were sensitive to the physical and social environments of the treatment student rooms.

Generally, women in both demographic groups, who had greater financial means or the support of their network, and who accessed PAC in private health facilities reported a better experience with the cleanliness of the premises. Women accessing care appreciated not only the state of the premises, but also the discretion of the place, as stated by this young woman:

> "*His (the private health provider's) clinic is clean and discreet…*" 19, female, single, student.

These statements were confirmed by this participant, a 30-year-old woman, married and civil servant:

> "*I truly did not have any problems at the clinic there…I truly appreciated the privacy at the clinic.*"

Some women who had sought care in public PAC services also found the premises clean enough.

> "*On that point (regarding the cleanliness of the rooms), the healthcare facility was clean.*" 36, female, married, housewife.

However, most participants reported that public facilities were not always clean and engaging.

> "*The room was not clean because it smelled of blood everywhere; it was in the delivery room that everything happened…I did not like the environment....*" 20, female, single, student.

Participants from both demographic groups were particularly concerned about the lack of privacy during the PAC treatment in public facilities, where the delivery room is typically the location for this procedure. The communal nature of these services increases the likelihood of patients encountering others from their community, which can result in the secret of their PAC treatment becoming the subject of gossip.

> "*I wanted to be in a corner without anyone knowing what had happened because when you come out, everyone is looking at you and knows what you have done, or they will whisper to determine what happened because they cannot see you with a baby.*" 20, female, single, student.

Another participant added the following:

> "*For the room, there was no problem. The real problem is the people around, it is very shameful, and you want to disappear even; there were so many people that, afterwards, I was obliged to travel; otherwise, it was not easy.*" 23, female, single, student.

However, some women who had connections in public health facilities were cared for in a discreet but not always comfortable manner in consultation rooms, as one participant reported:

> "*The maternity has its own building, but for me, it is in the infirmary that we did it, in an office (consultation room)... It is not simple. You walk into his (the nurse's) office while everyone is looking at you. (When) you are in pain, but you cannot cry loud for fear people will hear you; it was just pity.*" 24, female, single, student.

In contrast to the disparities mentioned earlier, the perceptions of women regarding the acceptability of the costs of PAC they received were rather similar across private and public services. Generally, women in both types of facilities found the high care costs. Despite PAC in public health facilities in Burkina Faso being subsidized, women often have to purchase drugs and supplies that are out of stock.

> "*Well, some products we had on site at the CSPS, but… all the products were not available there; we had to look for them at the pharmacy outside.*" 35, female, married, housewife.

Women who received PAC in private facilities also complained about the cost of drugs, which made the care even more expensive.

> "*They are a bit expensive, for the complication, I paid 30,000 CFA XOF (50 USD) more in addition to the 35,000 XOF (58) I had given so that he could give me the tablets.*" 19, female, single, student.

### "I received insults from the health workers": Competent care and abuse

Most participants viewed the providers as clinically competent, particularly given the effectiveness of treatment and resolution of abortion life-threatening complications. No patient had a choice between different treatment alternatives.

Nonetheless, a substantial number of women received counseling on family planning as part of PAC, and some reported having adopted modern contraceptive methods.

*"The health worker is competent; when he did the treatment in 30 minutes, I was relieved."* 19, female, single, informal worker.

Participant narratives suggested that healthcare provider attitudes may differ across facility types and vary with patient marital status.

Many women, regardless of marital status, who accessed PAC through private facilities described their experience as more respectful and dignified compared to those who sought care in public settings.

*"For me, it was not difficult because he was the same one who helped me have an abortion; he told me that if I had a problem, to come back and see him... No, I did not experience any delays."* 30, female, married, civil servant.

The same was true for this adolescent girl, whose boyfriend provided her with necessary money. The patient was referred to a private clinic for PAC.

*"My boyfriend provided me with the money for care... I had no difficulty getting care... I was not late for care because I called before coming, and when I arrived, he received me after another one with complications. He examined me quickly because he had given me the product. He truly cared about me."* 19, female, single, student.

In contrast, health workers in public facilities appeared more empathetic and understanding towards married women with complications resulting from an induced abortion. One woman who had a connection at the health facility for PAC shared that she received timely care.

*"When I arrived, the doctor was with a patient inside, and when he finished, he received me straight away and asked me what was wrong, I explained to him, then he examined me, and he prescribed me some products to take and to keep him informed of the rest."* 35, female, widowed, housewife.

However, some women in this group experienced judgmental attitudes from providers, as reported by this woman.

*"There was no respect; some of them even called me names; they called me names when they asked where my husband was and my sisters said he could not come there; some of them even went so far as to say that I was probably an unfaithful woman, and that is the consequence."* 33, female, married, housewife.

The experiences of single adolescents and young women receiving PAC in public facilities appeared to be at the discretion of the health worker responsible for their care. Many reported facing verbal abuse and public insults, which violated the confidentiality of their care and exposed them to stigma, neglect, and mistreatment.

*"When the health workers came to examine me after a long wait, they prescribed some medicines: tablets and blood samples. The relationship between the health workers and I was not favorable because I received insults from the health workers, and even said that it was because they knew my father; otherwise, they were going to report me to the local authorities."* 21, female, single, student.

One of the respondents wondered whether the provider intentionally inflicted painful treatment on her to punish her for her actions.

*"During the curettage, she did not play with me; she only scraped… I do not know if she did it with malice, but it was painful in a way!"* 20, female, single, student.

Some of these young women often reported understanding these mistreatments that they attributed to their own unworthiness and were surprised when healthcare providers showed them kindness.

*"He looked after me well, despite the crime I had committed…I didn't hear from anyone that I had an abortion."* 22, female, single, informal worker.

*"She respected us without insults, she gave us advice, and when someone came to ask what I was suffering from, she said that I had a stomachache to hide what my friend and I had done. She advised me to take [contraceptives]."* 20, female, single, student.

Some young women sometimes felt that some healthcare providers attitude was more an expression of their concern for their well-being rather than a moral judgement.

*"It's true that the health worker was hard at the beginning, but his advice truly helped me to recover better... Because of his advice, I came back to have an implant."* 19, female, single, informal worker.

*"For the care provided, there is nothing to say; she treated me like everyone else. Only when I remember her words: 'Look, you're going to kill yourself, you're so young and so pretty! don't do that anymore.' I just cry."* 24, female, married, student.

*"They did not make me feel ashamed, but they had pity for me. She told me not to repeat that (an induced abortion); that I was risking my life, so I should protect myself (with contraceptives) to avoid that."* 20, female, single, student.

### Patient satisfaction and fulfillment of human rights

Women's satisfaction with care is influenced by their experiences. Although most women reported satisfaction with PAC, adolescent girls and young women showed little satisfaction with the care they received. Participants mentioned feelings of powerlessness in their experiences.

*"I did not like the behavior of the health workers toward me. However, what can I say? In any case, I was not satisfied."* 18, female, single, student.

Participants, particularly adolescents and young women, had in general low expectations for proper quality PAC due to the high level of stigma surrounding abortion in their setting. Many respondents believed that being spared from death and criminal harassment was enough to feel satisfied. A young woman emphasized that in these words:

*"I was very satisfied; I can say that if I live today, it is thanks to them."* 21, female, single, student.

One woman agreed in the same way.

*"I was truly satisfied with the way they (the health workers) worked and the respect they gave me because I thought they were going to treat me badly or judge me for my choice, but this was not the case. "* 32, female, married, informal worker.

Despite their low expectations of the quality of care provided due to the reason for their visit, other respondents expressed a desire to be better understood and treated by health workers.

These findings reflect the diverse and intersecting factors shaping women's experiences with abortion and PAC. Building on the typologies previously described in Table 3, we present a conceptual diagram (Fig 2) to further illustrate how sociodemographic characteristics interact with abortion motivations, perceived stigma, and care-seeking patterns. This figure visually synthesizes the relationships between women's profiles, reasons for abortion, provider attitudes, and facility choice, highlighting how structural and social factors jointly shape accessibility and acceptability of care in this context.

## Discussion

This qualitative study explored women's experiences and perceptions of the accessibility and acceptability/person-centeredness of PAC in a small town in Burkina Faso. The results revealed that induced abortions were strongly disapproved in the community, with both association representatives and women who had an abortion sharing this view. Participant narratives reflected two predominant profiles among abortion seekers: young unmarried women who had an abortion because their partners did not recognize the pregnancy and risked societal condemnation, and married women who felt they were saving the lives of their children who were already born. Participants' experiences suggested that abortion stigma may contribute to delays or barriers in accessing PAC, particularly for unmarried women, which affects women's access to care, especially for those using public facilities. Regarding acceptance, young unmarried women more frequently described experiences of neglect or mistreatment by health workers, particularly in public facilities. However, all women declared themselves satisfied with the treatment received as their lives were saved. Stigmatizing social norms shaped the acceptability/person-centeredness of PAC, as women whose sexuality and abortions were particularly condemned tended to settle for low-quality care.

Taken together, our results showed that the women's perspective on the quality of care was strongly linked to the vision of abortion prevailing in their community. Culturally shaped ideas about who is more legitimate about having an induced abortion (in particular, married versus unmarried women) may affect equitable access and acceptability/person-centeredness of PAC for women in Kaya.

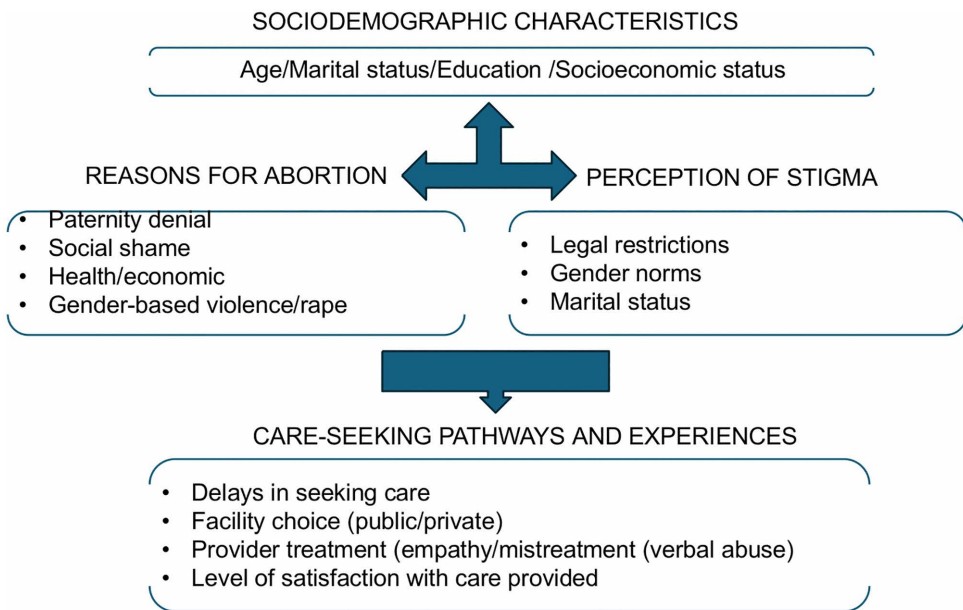

**Fig 2. Conceptual diagram of the relationship between sociodemographics characteristics, reasons for abortion and care seeking patterns.**

First, stigma, fear that their act will become known in the community, or that they will be reported for their crime strongly influences accessibility, particularly by delaying the seeking of health services for the PAC among the respondents. A study in Ghana in 2021 reporting on women's experiences of PAC who sought care in a regional hospital revealed that stigma and poverty delayed early access to services [37]. Adolescents and unmarried women in particular would only seek care when they were left with no other choice due to the severity of pain or complications. They also reported more delays in management by health workers, who might have been influenced by broader social and religious norms toward premarital sexuality. These findings are corroborated by a study in 2021 by Jacobson et al. in four countries, including two sub-Saharan African countries with more liberal legal frameworks, Ethiopia and Nigeria, but where abortion stigma remains high, highlighting that young people face social and structural barriers when accessing abortion and PAC [38].

The cultural norms surrounding abortion can impact a woman's perception of the care she receives. In our study, these dimensions of patient-centered care were important to women seeking PAC, but they had low expectations due to the hostile social climate surrounding abortion. Women sought discretion in care, which includes both the behavior of health workers and the organization of care services. In our study, young unmarried women were more likely to face disrespectful or indiscreet treatment from health workers. Previous research in stigmatized contexts, even in legal settings, has reported similar findings [8–10,39,40]. Our study found that women's experiences with PAC varied based on their characteristics, with adolescents and low-income single women more likely to report poor experiences. These findings align with previous studies that found that young single women, women of low socioeconomic status, and economically dependent women were more likely to report negative experiences. A study by Govule et al. in 2021 also found that living in West or Central Africa, rather than East Africa, was associated with worse care for these women care [41].

However, the women in our sample also reported promising trends in their experiences with PAC. Hence, irrespective of their demographic status (unmarried or married), participants in this study largely reported having been informed, counseled, and offered modern contraceptives during PAC to avoid being in the same situation. These reports of counseling and contraceptive provision indicate areas of potential strength within PAC that may warrant further support, as family planning is an important component of PAC quality. Evidence-based information provision, counseling, and contraception are key strategies for reducing maternal morbidity and mortality, especially in countries with restrictive abortion laws. The utilization of postabortion family planning, which is the root cause of induced abortion, is critical for reducing high levels of unintended pregnancy. PAC services may offer a strategic entry point for expanding youth access to family planning and for women who miss an opportunity during their postpartum care [42–44].

Our findings showed that, regardless of their characteristics, women were satisfied once the treatment they received allowed them to preserve their lives. Although some respondents asked for more consideration during care, most felt grateful for having been saved, a favor that the care staff granted them. Eboigbe et al., in 2022, in a secondary analysis of a multicountry study in SSA, showed that the majority of both adolescents and older women reported relatively high levels (at least 75%) of satisfaction when examining each question related to a dimension of satisfaction, with the exception of the amount paid for "out-of-pocket" services [45]. Hence, their satisfaction was not directly related to the quality of the PAC service itself but rather to getting through a difficult situation. Previous studies that reported similar findings argued that there was a potential relationship between patients' expectations of low-quality care, which stemmed from social norms and stigma, and their perceptions of the quality of care received [21,8,9].

This study showed that most women were unaware that abortion was allowed in certain situations, and only a small number of association representatives were knowledgeable about this. This finding is consistent with a 2015 systematic review, which found that women's knowledge of abortion laws, legal grounds, and restrictions was limited, even in countries like Ghana and Zambia where the laws are more permissive. For example, in Ghana in 2017, only 11% of women of reproductive age who were familiar with abortion knew the legal grounds on which it was legal in the country. Additionally, the review found that poorer and less educated women had lower levels of knowledge. A lack of information disclosure to the general public, healthcare providers, and other stakeholders is a common obstacle in many countries that have

undergone abortion law reform and has hindered the implementation of these policies. The stigma surrounding abortion may contribute to this reluctance to speak out about existing legal grounds. Withholding information could be a means of avoiding backlash from community leaders who are hostile to the issue, or it could serve as a means of controlling access to abortions, even when they are legal.

Our findings offer valuable insights for policymakers, mid-level managers, healthcare providers, and researchers in designing, monitoring, or assessing the implementation of safe abortion or PAC policies. The narratives of women regarding their experiences in accessing PAC in Kaya highlight the significant influence of stigma, social identity, and provider discretion on both their expectations and experiences of care. This is particularly evident among adolescents and young unmarried women. Many of these individuals expressed relief at avoiding humiliation or legal repercussions rather than satisfaction with the quality of care itself, thereby highlighting diminished expectations that reflect broader systemic inequities. To ensure equitable access and person-centered treatment, PAC strategies must be implemented at multiple levels. National health policies should promote consistent standards of respectful and rights-based care to enhance the quality of PAC, particularly in Kaya and similar settings, and more broadly across the entire country. This approach aims to prevent variations in care quality based on marital status, socioeconomic background, or geographic location. Furthermore, improvements at the facility level, including adequate infrastructure, privacy, and youth-sensitive provider training, are crucial for increasing acceptability and restoring patient agency. At the community level, limited awareness of abortion laws and strong moral condemnation continue to delay care-seeking and reinforce stigma. Enhancing public education initiatives on reproductive rights and the availability of PAC, as well as engaging trusted local actors to promote open dialogue, can contribute to altering detrimental narratives and improving community support for PAC services. Lastly, targeted strategies for adolescents and young people are essential. Strengthening youth-friendly services and integrating comprehensive sexuality education into school curricula can help improve knowledge, reduce stigma, and facilitate timely and dignified access to care.

Our study had several limitations. Hence, because the study was limited to one specific area and the sample size was relatively small, we cannot generalize the findings to the entire country, even within rural areas. The lack of systematic data prevented us from studying the role of socioeconomic resources in accessing acceptable PAC in detail. In addition, the patients were asked to provide their answers about an experience that occurred in the three years preceding the interview; the data could be affected by some recall bias. Given the sensitive nature of the topic and participants' refusal to be recorded, data collection relied on real-time note-taking during interviews. While interviewers expanded field notes immediately afterward to ensure completeness and context, this method carries inherent risks, including incomplete transcription and reduced verbatim accuracy. We also acknowledge the potential for desirability bias, as abortion remains socially disapproved in the study setting. Some respondents may have shaped their narratives to conform to normative expectations, despite assurances of confidentiality. Furthermore, translation from the local language to French first, and then into English may have resulted in the loss of linguistic nuance, despite the lead author's bilingual proficiency and collaborative debriefings with the research team. These limitations may have influenced the precision of participant accounts and thematic interpretation. Also, while the use of Braun and Clarke's reflexive thematic analysis allowed for interpretive depth within a single-coder framework, we acknowledge the limitations inherent in relying on one researcher for coding. Although the lead author is bilingual in Moore and French and familiar with the study context, the absence of intercoder reliability assessments may affect reproducibility. In addition, we acknowledge that dividing participants into specific age groups and marital status categories may introduce bias, especially given the limited sample size and qualitative nature of the study. These classifications were used descriptively to examine recurring patterns in women's narratives but do not reflect rigid or causal associations. Experiences of stigma, access, and provider interactions are inherently nuanced, and we caution against generalizing these groupings beyond the context of this study. Finally, we did not collect facility-level type or capacity data beyond the public/private categorization and thus were unable to stratify women's accessibility and acceptability experiences by CSPS, CMA, or

CHR. Future research should incorporate facility-level capacity measures to examine how service scope and infrastructure shape women's postabortion care experiences in this setting.

Despite these limitations, our study presents some interesting findings that contribute to the understanding of the quality of PAC from the patient's perspective. First, our approach (respondent-driven sampling) allowed us to obtain information at the community level from the population of abortion seekers in a secondary city in Burkina Faso, which is difficult to identify because of the stigma surrounding the practice. This could reduce the desirability bias often induced when interviews are conducted in health facilities. In addition, by exploring both the accessibility and acceptability/person centeredness of PAC care from the perspective of the beneficiaries and understanding their variation in relation to two types of abortions that emerged as stigmatized differently in the community (among unmarried and married individuals), we contribute to a better understanding of the link between social and cultural contexts and the quality of PAC.

## Conclusion

This study highlighted inequalities in the acceptability/person centeredness and accessibility of PAC received by women, according to their sociodemographic status. Various sociocultural, moral, and economic antecedents shape people's expectations of health care and subsequently influence access to and acceptability/person-centeredness of the care they receive. The aspects of quality prioritized by these women in an abortion service were context specific and greatly influenced by the prevailing abortion stigma, which was stronger for unmarried women. It is important to consider and respect the needs and concerns of women to ensure that the services offered are of high quality and meet their expectations, leading to greater patient 'satisfaction, thereby enabling equitable access to timely and optimal quality care and the continued sustainable use of health services for better health. Particular attention should be paid to how to address differential stigma when designing and planning a safe abortion care program.

## Supporting information

**S1 File. Standard for Reporting Qualitative Research (SRQR).**
(DOCX)

**S1 Checklist. Checklist.**
(DOCX)

## Acknowledgments

We gratefully acknowledge the invaluable contributions of the field team, ACLPK, BC, PAJT and MS, whose dedication to data collection and respectful engagement with participants were instrumental to the success of this study.

## Author contributions

**Conceptualization:** Rachidatou Compaoré, Clementine Rossier, Seni Kouanda.

**Data curation:** Rachidatou Compaoré, Clementine Rossier.

**Formal analysis:** Rachidatou Compaoré, Seni Kouanda.

**Funding acquisition:** Rachidatou Compaoré, Seni Kouanda.

**Investigation:** Rachidatou Compaoré, Adama Baguiya, Nazi Vincent Bagnoa, Seni Kouanda.

**Methodology:** Rachidatou Compaoré, Clementine Rossier, Adama Baguiya, Seni Kouanda.

**Project administration:** Rachidatou Compaoré, Seni Kouanda.

**Resources:** Rachidatou Compaoré, Seni Kouanda.

**Software:** Rachidatou Compaoré.

**Supervision:** Rachidatou Compaoré, Adama Baguiya, Nazi Vincent Bagnoa, Seni Kouanda.

**Validation:** Rachidatou Compaoré, Clementine Rossier, Seni Kouanda.

**Visualization:** Rachidatou Compaoré.

**Writing – original draft:** Rachidatou Compaoré, Clementine Rossier, Seni Kouanda.

**Writing – review & editing:** Rachidatou Compaoré, Clementine Rossier, Onikepe Owolabi, Adama Baguiya, Caron Kim, Moussa Zan, Nazi Vincent Bagnoa, Martin Bangha, Seni Kouanda.

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
