## [Decision Letter · Decision Letter 0]

16 Jan 2025

PGPH-D-24-02701

"As it is about that, they do as they please": Women’s experience of the accessibility and acceptability of postabortion care in Kaya, Burkina Faso

Dear Dr. Compaoré,

Thank you for submitting your manuscript to PLOS Global Public Health. After careful consideration, we feel that it has merit but does not fully meet PLOS Global Public Health’s publication criteria as it currently stands. Therefore, we invite you to submit a revised version of the manuscript that addresses the points raised during the review process.

Please address the reviewer's comments

We look forward to receiving your revised manuscript.

Kind regards,

Ejemai Eboreime, MD, MSc, PhD

Academic Editor

Journal Requirements:

1. Please provide an Author Summary. This should appear in your manuscript between the Abstract (if applicable) and the Introduction, and should be 150–200 words long. The aim should be to make your findings accessible to a wide audience that includes both scientists and non-scientists. Sample summaries can be found on our website under Submission Guidelines:

https://journals.plos.org/globalpublichealth/s/submission-guidelines#loc-parts-of-a-submission.

2. In the online submission form, you indicated that "The data supporting the findings of this study are available from the corresponding author upon reasonable request.". 

a. In a public repository, 

b. Within the manuscript itself, or 

c. Uploaded as supplementary information.

Additional Editor Comments (if provided):

Reviewers' comments:

Reviewer's Responses to Questions

**Comments to the Author**

1. Does this manuscript meet PLOS Global Public Health’s publication criteria ? Is the manuscript technically sound, and do the data support the conclusions? The manuscript must describe methodologically and ethically rigorous research with conclusions that are appropriately drawn based on the data presented.

Reviewer #1: Partly

Reviewer #2: Yes

2. Has the statistical analysis been performed appropriately and rigorously?

Reviewer #1: N/A

Reviewer #2: Yes

3. Have the authors made all data underlying the findings in their manuscript fully available (please refer to the Data Availability Statement at the start of the manuscript PDF file)?

Reviewer #1: No

Reviewer #2: Yes

4. Is the manuscript presented in an intelligible fashion and written in standard English?

Reviewer #1: Yes

Reviewer #2: Yes

5. Review Comments to the Author

Reviewer #1: Abstract

• Clarify what types of associations the representatives are from. Are they activism groups, medical professionals, grassroots organizations, governmental bodies, or other entities?

Introduction

• Line 119-124: “Approximately 90% of abortions are unsafe, and there is evidence of disparities among older women, women with less education, and women living in rural areas [31]… However, despite these policies, almost 40% of women who experience abortion-related complications do not receive the necessary medical care [31].” Presenting these data, which are from Argentina, Bangladesh, Ethiopia, and Nigeria, might mislead readers into thinking that the data belongs to Burkina Faso. Do you have local data available for these measures?

• Line 141: “Our data collection tool incorporated questions from both the WHO framework for quality of care [22] and the abortion stigma literature.” Consider adding a citation for the abortion stigma literature.

• Line 150: “Contributing to the achievement of sustainable development goals (SDGs).” It would be helpful to specify which SDGs your study contributes to.

Materials and Methods

• Line 160-166: “The Kaya health district includes forty (40) first-level public health facilities, including 39 primary health and social promotion centers (CSPSs) and one medical center with a surgery unit (CMA). In addition, there were four private and confessional health facilities. The health district includes the regional hospital of Kaya (CHR), which is a referral hospital for the Centre-north region. In 2020, 53 obstetric complications due to unsafe abortions were reported in the Kaya Health District, representing 8.3% of all obstetric complications (in the country) [16].” What are the capacities of these health facilities? Do they all provide PAC?

• I would like to know more about how the interview guides overtime based on the emerging themes. If possible, please detail the process and present your different versions of question guides and final codebook for both types of participants.

• Include the length of the interviews in the Results section.

• Regarding the note-taking process during interviews: Since voice recording was not possible due to sensitivity concerns, did researchers take real-time transcripts in English, Moore, or French? Were the transcripts and field notes written simultaneously? Did the interviewers reflect on the interviews afterward? Was there a translation process involved? Who coded the transcripts, and what are their language proficiencies? Were they acknowledged in the author list?

• Clarify whether each participant was interviewed by a single researcher or if additional personnel assisted with field notes. Recognize the limitations of your methods in transcribing, note-taking, and translation.

• Elaborate on whether the authors discussed findings with the data collection and coding teams. How were discrepancies among coders resolved? What was the level of agreement between coders?

• While you attached the SRQR, there are still ambiguities in your data analysis methods. Please provide further detailed clarification.

Ethical Considerations

• Since women below 18 were included, how was their consent obtained—through parental consent or another method?

• Why was only verbal informed consent obtained? Were there risks associated with obtaining written consent? What measures were taken to ensure participants understood the risks involved in the study?

Results

• Ensure consistency in how participant information is presented, including age, gender, marital status, and professional titles.

• Consider moving the section on “sociodemographic characteristics of abortion seekers and reasons for abortion” and Table 1 to the beginning of the Results section. This reorganization would provide a clearer overview of the participants and improve the flow of your analysis.

• Line 483: “As it is about that, they do as they please”—This phrase appears in both the title and subtitle but is somewhat ambiguous. It was unclear initially whether it represents the perspective of the women or the care providers. In the Results section, this phrase is linked to the acceptability/person-centeredness sub-section but is only discussed later under “Patient satisfaction and fulfillment of human rights.” Consider clarifying how you structured the Results section and whether this quote should be emphasized as a key finding.

• Since your analysis compares perspectives from younger single women and older married women, creating a table to summarize the differences and similarities between these groups would greatly enhance clarity. It will be great if you can also visualize your findings by creating a conceptual diagrams.

• After introducing the abbreviation for PAC, avoid repeating “postabortion care (PAC)”.

Discussion

• Line 622: Consider rephrasing “Clients” for clarity.

• Line 736-747: Expand on the study's implications for both patients and communities.

• Acknowledge the limitations of your transcribing, note-taking, and translation methods, as noted earlier.

• Since this is a qualitative study with a relatively small sample size, dividing women into specific age bucket and marital status categories might introduce bias. Acknowledge this limitation and avoid using potentially causal languages throughout the article.

Reviewer #2: The abstract explores the challenges faced by women in accessing postabortion care (PAC) in Kaya, Burkina Faso, emphasizing the impact of legal restrictions and social stigma, particularly for impoverished rural women. Using in-depth interviews with 35 women and 8 association representatives, the study identifies inequalities in PAC access and acceptability, shaped by women’s sociodemographic status. Findings reveal that abortion stigma is stronger for unmarried women, delaying care-seeking and leading to verbal abuse, longer waits, and lack of privacy, whereas married women face less stigma due to socially accepted reasons for abortions. Socioeconomic disparities also influence care, with wealthier women accessing better-quality private facilities. The study highlights significant inequalities and calls attention to the need for policies addressing the stigma, prosecution fears, and disparities in PAC provision.

6. PLOS authors have the option to publish the peer review history of their article (what does this mean? ). If published, this will include your full peer review and any attached files.

**Do you want your identity to be public for this peer review?** For information about this choice, including consent withdrawal, please see our Privacy Policy .

Reviewer #1: No

Reviewer #2: No

---

## [Decision Letter · Decision Letter 1]

18 Sep 2025

PGPH-D-24-02701R1

"As it is about that, they do as they please": Women’s experience of the accessibility and acceptability of postabortion care in Kaya, Burkina Faso

Dear Dr. Compaoré,

Thank you for submitting your manuscript to PLOS Global Public Health. After careful consideration, we feel that it has merit but does not fully meet PLOS Global Public Health’s publication criteria as it currently stands. Therefore, we invite you to submit a revised version of the manuscript that addresses the points raised during the review process.

We look forward to receiving your revised manuscript.

Kind regards,

Ejemai Eboreime, MD, MSc, PhD

Academic Editor

Journal Requirements:

Additional Editor Comments (if provided):

Reviewer #1:

Reviewers' comments:

Reviewer's Responses to Questions

**Comments to the Author**

1. If the authors have adequately addressed your comments raised in a previous round of review and you feel that this manuscript is now acceptable for publication, you may indicate that here to bypass the “Comments to the Author” section, enter your conflict of interest statement in the “Confidential to Editor” section, and submit your "Accept" recommendation.

Reviewer #1: (No Response)

2. Does this manuscript meet PLOS Global Public Health’s publication criteria ? Is the manuscript technically sound, and do the data support the conclusions? The manuscript must describe methodologically and ethically rigorous research with conclusions that are appropriately drawn based on the data presented.

Reviewer #1: Yes

3. Has the statistical analysis been performed appropriately and rigorously?

Reviewer #1: N/A

4. Have the authors made all data underlying the findings in their manuscript fully available (please refer to the Data Availability Statement at the start of the manuscript PDF file)?

Reviewer #1: No

5. Is the manuscript presented in an intelligible fashion and written in standard English?

Reviewer #1: Yes

6. Review Comments to the Author

Reviewer #1: I am honored to review this manuscript, which is clearly written with strong logic and professional academic style. It was a pleasure to review, and I would like to congratulate the authors on completing such critical and meaningful public health work in a rural setting.

I have a few suggestions for strengthening the manuscript:

1. The authors note that “The Kaya health district includes forty (40) first-level public health facilities, including primary health and social promotion centers (CSPSs) and one medical center with a surgery unit (CMA). In addition, there were four private and confessional health facilities. The health district includes the regional hospital of Kaya (CHR).” However, in Table 1 only private vs. public facilities are shown. Do we know which specific types/levels of health facilities women visited, and the capacity of those facilities? Stratifying women’s experiences by facility type could add depth and could be reflected either in the table or in the text.

2. The manuscript mentions selecting eight community representatives from grassroots organizations. Why were these particular individuals chosen? Why not include providers, given that the study aims to explore tension and interpersonal interactions in care settings (which could be a limitation)? Also, what are their demographics? This information would be better placed in a separate table rather than alongside women’s characteristics.

3. Line 239: the length of the interviews belongs in the results section.

4. Lines 310–312: the authors state, “we developed two typologies based on women’s socioeconomic status and the degree of stigma surrounding their abortion,” which is excellent. However, I would like to see a clearer connection between socioeconomic status and degree of stigma illustrated in a visual diagram. Table 2 is strong, but it omits economic status, and it does not fully capture the interactions between dimensions that are partially reflected in Figure 2. The authors may benefit from creating a more comprehensive figure to tease out these nuances, or from adding a paragraph explicitly connecting Table 2 and Figure 2.

5. Lines 392–402 and 337–346: it would improve readability to provide a comprehensive overview of respondent characteristics at the start of the results section, combining these two parts.

7. PLOS authors have the option to publish the peer review history of their article (what does this mean? ). If published, this will include your full peer review and any attached files.

**Do you want your identity to be public for this peer review?** For information about this choice, including consent withdrawal, please see our Privacy Policy .

Reviewer #1: No

---

## [Decision Letter · Decision Letter 2]

27 Nov 2025

PGPH-D-24-02701R2

"As it is about that, they do as they please": Women’s experience of the accessibility and acceptability of postabortion care in Kaya, Burkina Faso

Dear Dr. Compaoré,

Thank you for submitting your manuscript to PLOS Global Public Health. After careful consideration, we feel that it has merit but does not fully meet PLOS Global Public Health’s publication criteria as it currently stands. Therefore, we invite you to submit a revised version of the manuscript that addresses the points raised during the review process.

We look forward to receiving your revised manuscript.

Kind regards,

Helen Howard

Staff Editor

Journal Requirements:

Additional Editor Comments (if provided):

Reviewers' comments:

Reviewer's Responses to Questions

**Comments to the Author**

1. If the authors have adequately addressed your comments raised in a previous round of review and you feel that this manuscript is now acceptable for publication, you may indicate that here to bypass the “Comments to the Author” section, enter your conflict of interest statement in the “Confidential to Editor” section, and submit your "Accept" recommendation.

Reviewer #1: (No Response)

2. Does this manuscript meet PLOS Global Public Health’s publication criteria ? Is the manuscript technically sound, and do the data support the conclusions? The manuscript must describe methodologically and ethically rigorous research with conclusions that are appropriately drawn based on the data presented.

Reviewer #1: Yes

3. Has the statistical analysis been performed appropriately and rigorously?

Reviewer #1: N/A

4. Have the authors made all data underlying the findings in their manuscript fully available (please refer to the Data Availability Statement at the start of the manuscript PDF file)?

Reviewer #1: No

5. Is the manuscript presented in an intelligible fashion and written in standard English?

Reviewer #1: Yes

6. Review Comments to the Author

Reviewer #1: The revision looks strong. I have just a few minor comments. In Table 1, it would help to clarify the distinction between “informal” and “formal” professions—this could be addressed with a brief footnote. Also in Table 1, the “number of children” category lists 0, whereas page 16 lists 1 and “+” under socio-demographics; this appears to be an inconsistency that can be easily corrected. Once these small issues are addressed, I see no concerns with the article moving forward to publication in your journal.

7. PLOS authors have the option to publish the peer review history of their article (what does this mean? ). If published, this will include your full peer review and any attached files.

**Do you want your identity to be public for this peer review?** For information about this choice, including consent withdrawal, please see our Privacy Policy .

Reviewer #1: No

Figure Resubmissions:While revising your submission, we strongly recommend that you use PLOS’s NAAS tool (https://ngplosjournals.pagemajik.ai/artanalysis) to test your figure files. NAAS can convert your figure files to the TIFF file type and meet basic requirements (such as print size, resolution), or provide you with a report on issues that do not meet our requirements and that NAAS cannot fix.

---

## [Editor Report · Decision Letter 3]

16 Dec 2025

"As it is about that, they do as they please": Women’s experience of the accessibility and acceptability of postabortion care in Kaya, Burkina Faso

PGPH-D-24-02701R3

Dear Dr. Compaoré,

We are pleased to inform you that your manuscript '"As it is about that, they do as they please": Women’s experience of the accessibility and acceptability of postabortion care in Kaya, Burkina Faso' has been provisionally accepted for publication in PLOS Global Public Health.

Best regards,

Julia Robinson

Executive Editor